# A Negative Regulatory Role for RKIP in Breast Cancer Immune Response

**DOI:** 10.3390/cancers14153605

**Published:** 2022-07-24

**Authors:** Vu N. Bach, Jane Ding, Miranda Yeung, Taylor Conrad, Hussain N. Odeh, Paige Cubberly, Christopher Figy, Han-Fei Ding, Robert Trumbly, Kam C. Yeung

**Affiliations:** 1Department of Cell & Cancer Biology, College of Medicine and Life Sciences, University of Toledo, Health Science Campus, Toledo, OH 43614, USA; vbach@rockets.utoledo.edu (V.N.B.); miranda.yeung@utoledo.edu (M.Y.); taylor.conrad@utoledo.edu (T.C.); hussain.odeh@rockets.utoledo.edu (H.N.O.); paige.cubberly@rockets.utoledo.edu (P.C.); christopher.figy@rockets.utoledo.edu (C.F.); 2Georgia Cancer Center, Medical College of Georgia, Augusta University, Georgia, GA 30912, USA; jding@augusta.edu (J.D.); hding@augusta.edu (H.-F.D.); 3Department of Medical Education, College of Medicine and Life Sciences, University of Toledo, Health Science Campus, Toledo, OH 43614, USA

**Keywords:** RKIP, immunomodulator, cytokines, interferons

## Abstract

**Simple Summary:**

Breast cancer is the second most common cancer in women worldwide. Regulation of breast cancer metastasis remains an elusive phenomenon. Elucidating the mechanistic pathway of metastatic signaling may identify targets for regulating cancer metastatic potential. Raf-1 kinase inhibitor protein (RKIP) has been shown to negatively regulate signaling pathways involved in cancer progression and metastasis. RKIP may suppress metastasis of breast cancer cells by downregulating elements of the immune system.

**Abstract:**

Raf-1 kinase inhibitor protein was first identified as a negative regulator of the Raf signaling pathway. Subsequently, it was shown to have a causal role in containing cancer progression and metastasis. Early studies suggested that RKIP blocks cancer progression by inhibiting the Raf-1 pathway. However, it is not clear if the RKIP tumor and metastasis suppression function involve other targets. In addition to the Raf signaling pathway, RKIP has been found to modulate several other signaling pathways, affecting diverse biological functions including immune response. Recent advances in medicine have identified both positive and negative roles of immune response in cancer initiation, progression and metastasis. It is possible that one way that RKIP exerts its effect on cancer is by targeting an immune response mechanism. Here, we provide evidence supporting the causal role of tumor and metastasis suppressor RKIP in downregulating signaling pathways involved with immune response in breast cancer cells and discuss its potential ramification on cancer therapy.

## 1. Introduction

RKIP was first identified as the binding partner of Raf-1 kinase [1]. Raf-1 kinase is the activating kinase of the MEK-Erk kinase cascade that functions downstream of the membrane-associated Ras small GTPase. Upon stimulation by growth factors, GTP-bound Ras activates Raf-1. The Raf-MEK-Erk module constitutes the core mitogen signaling pathway that governs the proliferation, survival and differentiation of many different cell types [2,3]. RKIP is identical to a previously described 23 kDa phosphatidylethanolamine-binding protein (PEBP1) of the PEBP protein family. This family is a largely conserved group of proteins found in a variety of organisms, including bacteria, yeast, nematodes, plants, drosophila and mammals [4]. Due to the inhibitory effect of RKIP on Raf signaling, we hypothesized early on that it would have a profound regulatory role on cancer biology [1]. It was later shown with genetically engineered mouse models and cancer cells transplantation experiments that RKIP has a regulatory role in cancer progression and metastasis [5,6,7]. However, the restrictive effect on cancer progression caused by RKIP expression is not fully restored by a gain-of-function in the Erk1/2 pathway, indicating that there are additional RKIP targets [8,9].

Affinity chromatography experiments showed that RKIP bound to phospholipids and that it had a strong preference for phosphatidylethanolamine (PE), which is found predominantly in the inner side of the plasma membrane. It was shown that binding of PE enhanced the dissociation of Raf-1 from RKIP [10,11]. Subsequently RKIP, was found to interact with a great variety of proteins. The ability to interact with multiple proteins and to bind to phospholipids allows RKIP to modulate additional signaling pathways and biological functions [12,13,14,15,16,17]. Other known functions that are regulated by RKIP include mitotic checkpoint, immune response, cardiac function, and neuronal development [18,19,20,21,22,23,24,25,26]. It remains unclear if RKIP cancer suppressor and metastasis inhibition were also dependent on its effects on these other functions. In this study, we show that RKIP expression is associated with the repression of signaling pathways and genes involved in immune response in breast cancer cells.

## 2. Methods

### 2.1. Cell Lines and Reagents

168FARN breast cancer cell lines were kindly provided by Dr. Fred Miller (Karmanos Cancer Institute, Detroit, MI, USA), and MCF7 cells were purchased from ATCC. Cells were cultured as described [5].

### 2.2. Microarray

Total RNA was prepared with Trizol (Invitrogen) from RKIP or control knockdown 168FARN cells in three independent experiments [5]. The effect of downregulated RKIP expression on mouse transcriptome was examined with Affymetrix Mouse Gene 2.0 ST microarray chip. The Partek Genomics Suite was employed to calculate the fold change in expression with significance determined by ANOVA after the data were normalized. The raw data and expression values for the microarray experiment were deposited to NCBI GEO with the accession number GSE206259. Gene ontology enrichment was analyzed with the GSEA program [27] for all differentially expressed genes (R ± 1.5-fold, *p* < 0.01).

### 2.3. Analysis of Tumor Datasets

The tumor gene expression data were accessed at https://www.cbioportal.org/ (accessed on 12 April 2021). The Breast Invasive Carcinoma (TCGA, PanCancer Atlas) dataset was selected, consisting of 1084 samples. After choosing Query by gene, mRNA expression z-scores relative to all samples (log RNA Seq V2 RSEM) was selected. The default cutoff z-score of 2.0 was retained. PEBP1 (=RKIP) was entered as the query gene. After the OncoPrint diagram was displayed, the Coexpression tab was selected. After the table was displayed, download tsv was chosen, and this file was opened in Excel.

### 2.4. Quantitative Real-Time Reverse Transcriptase–Polymerase Chain Reaction (qRT-PCR)

qRT-PCR was performed as described [9]. Glyceraldehyde-3-phosphate dehydrogenase was used as an internal control. Sequences of the primers used in qRT-PCR will be available upon request.

## 3. Results

### 3.1. Gene Set Enrichment Analysis of Genes Affected by Reduced Expression of RKIP

To investigate if RKIP targets other signaling pathways and biological functions to interfere with cancer progression and metastasis, we analyzed the expression of ~27,000 protein encoding gene transcripts in the mouse breast cancer cell line 168FARN with or without downregulation of RKIP expression with specific shRNA by DNA microarray profiling. The cancer line 168FARN was chosen because of the existence of an allograft breast cancer model with this cell line for in vivo study and because of its similarity to human breast cancer triple negative subtype. We generated a ranked gene list according to fold-change affected by the knocking down of RKIP expression when compared to the control knockdown. The list was submitted to GSEA (Gene Set Enrichment Analysis) using the GSEA program. The Hallmark gene sets from MSigDB (Molecular Signature DataBase) were used to test for enrichment. Each gene set comprises genes that display coherent expression and represent specific well-defined biological states or processes. The Hallmark collection, which has only 50 gene sets, was chosen to identify the major pathways and to avoid the redundancy of the complete gene ontology sets. The GSEA produces lists of the gene sets enriched near the top of the ranked genes (upregulated) or near the bottom (downregulated) or both.

Results of GSEA (FDR < 0.09, *p* < 0.02) were visualized using the Enrichment Map plugin available for Cytoscape (Figure 1). Twenty gene sets were significantly enriched and linked by nine edges, while eleven gene sets were significantly upregulated in phenotype upregulated, and nine gene sets were significantly upregulated in phenotype downregulated. Gene sets involved in inflammation/immune response dominated, and all of them were upregulated in phenotype upregulated, showing high redundancy with commonly shared genes (Figure 2). Some of the shared genes include transcription factors (TF) IRF7, IRF9, IRF1, STAT1 and STAT4.

Genes affected by the RKIP knockdown with an FDR-q value of <0.1 were submitted for GSEA analysis using the MSigDB Transcription Factor Target collection. Leading edge analysis identified 12 motifs within 4 kb of the transcription start site for TF binding (Figure 3a). Consistently, the enriched predominant motifs are for the binding of interferon regulatory factors (IRFs) or a combination of IRF and STAT (ISRE) (Figure 3b,c). Our results therefore suggest that RKIP expression is associated with repression of immune response genes in cultured mouse breast cancer cells.

To validate the results from our DNA microarray analysis, we examined the effects of expression of a selected set of genes in RKIP knockdown 168FARN cells by qRT-PCR. In light of our bioinformatics analysis results, we focused on several well-studied interferon response genes for study. The selected genes include interferon genes themselves and the genes induced by them. We also included genes that encode factors regulating these responses. They are the type 1–3 interferons (IFNa/b, g, and l), chemokines and cytokines induced by IFNs (CCL2, CCL5, CCL7, CXCL10, IL6 and TFNa) and transcription factors that regulate interferon response pathways (IRF-3 and -7). Consistent with DNA microarray analyses results, we observed significant upregulation of all the selected interferon response genes upon silencing of RKIP expression in 168FARN cells (*p* < 0.05, unpaired Student’s *t* test) (Figure 3d). The negative regulatory effects of RKIP on interferon response genes is not mouse breast cancer cell-type specific, as we observed similar results with human luminal breast cancer cell line MCF7 (Figure 3e). Our qRT-PCR results therefore provide direct experimental proof of our DNA–microarray bioinformatics findings.

### 3.2. Gene Set Enrichment Analysis of Genes Correlate with Expression of RKIP in Human Breast Tumors

To determine clinical validity of our in vitro cell-based findings, we identified genes that were co-expressed with RKIP in human breast tumors. The gene expression data were accessed at https://www.cbioportal.org/ (accessed on 12 April 2021). A list of the correlations of expression of all human genes with RKIP was compiled and submitted to GSEA for analysis using Hallmark gene sets for the enrichment test. Results from the GSEA analysis of the RKIP correlation data are shown in Figure 4. Twenty-nine gene sets were significantly enriched and linked by 11 edges with FDR < 0.09 and *p* < 0.02. While 10 gene sets were significantly upregulated in phenotype upregulated, 20 gene sets were significantly upregulated in phenotype downregulated. Among the 29 gene sets, 14 of them overlap with gene sets enriched in the gene list generated with RKIP knockdown breast cancer cells cultured in vitro. Importantly, six of the overlapped gene sets are involved in immune response (Figure 5).

Direct comparisons of the gene sets found in the knockdown and correlation analyses are shown in Table 1 and Table 2. Table 1 shows the most significant categories for upregulated in RKIP knockdown and those negatively correlated with RKIP expression in breast cancer. The top nine categories down in the RKIP knockdown all match categories in the top 11 of the sets negatively correlated with RKIP expression. Table 2 shows the most significant categories for downregulated in RKIP knockdown and those positively correlated with RKIP expression in breast cancer. Five of the eight top categories up in the RKIP knockdown match categories in the top eight of the sets positively correlated with RKIP expression.

The mouse breast cancer cell line FARN168 is considered as a triple-negative subtype. We analyzed the co-expression of RKIP mRNA by breast cancer subtype (PAM50) using the same TCGA RNA-seq data as before (Appendix A). Appendix A shows the GSEA results for all samples and subtypes. The categories related to inflammation and immune response are highlighted in yellow. All subtypes show an enrichment for gene sets in these categories. Therefore, the relationship between RKIP and these pathways is not limited to a specific breast cancer subtype. The overlaps between the subtypes are visualized in the Venn diagrams of Appendix A.

Consistently, the gene sets that are downregulated in the knockdown experiment are strongly correlated with RKIP expression in breast cancer. This suggests that the effects of RKIP on the expression of genes involved with immune response are not species specific.

## 4. Discussion

Immune response is a defense mechanism that evolved to combat infection and tissue injury. It is generally divided into innate and adaptive immunity [28]. Innate immunity is our first line of defense and consists of myeloid cells, natural killer cells, and innate lymphoid cells. It reacts quickly to invading pathogens and damaged tissue by recognizing pathogen-associated molecular patterns (PAMPs) or host-derived damaged-associated molecular patterns (DAMPs). PAMPs and DAMPs are detected by a great variety of receptors known as PRRs (pattern recognition receptors), which are present on the cell surface and in the cytoplasm of innate immune cells. PRR recognition of PAMPs and DAMPs activates several transcription factors including NF-kB and IRFs. These transcription factors then increase the expression of genes encoding chemokines, cytokines, and enzymes that are crucial for inflammation initiation and leukocyte recruitment and activation. A subset of PRRs activate caspase-1, a protease that cleaves the pro-cytokines IL1b and IL18 to functional cytokines [29,30]. Adaptive immunity consists of B- and T-lymphocytes. They are slower to respond but do so specifically and create an immunological memory from the insults. This immunological memory is advantageous from an evolutionary standpoint, as it allows the body to remember previous insults and therefore mounts a more rapid and effective response [28].

Experiments with global RKIP knockout mice revealed a regulatory role of RKIP in the inflammatory response to pathogens and tissue injury. By interacting with multiple signaling molecules in both immune and non-immune cells, RKIP was demonstrated to have both retraining and supportive roles in inflammation initiated by innate and adaptive immune response. Notably, it was shown that the induction of the synthesis of cytokines, TNFα, IFNβ and IL6 in macrophages by viral infection was RKIP dependent [19]. RKIP was also shown to play a positive role in systemic inflammatory response syndrome in mice, which was caused by excessive production of IFN-γ by Vβ3+ T cells-stimulated splenocytes [31].

The molecular basis of cytokine induction by viral infection has been elucidated. Recognition of viral PAMPs by PRRs mostly activates the TANK-binding kinase 1 (TBK1), which then phosphorylates and activates the transcription factor–interferon regulatory factors 3 (IRF3) and 7 (IRF7), ensuing their homo- or heterodimerization. Dimerized IRFs translocate to the nucleus and stimulate expression of chemokines and cytokines, including small amounts of IFN-β and -α. Secreted IFN-β and -α bind to the cytokine dimeric receptor IFNAR1/2 in an autocrine and paracrine pattern to catalyze activation of phosphorylation of the receptors and receptor-bound STATs 1 and 2. Phosphorylated STAT1/2 heterodimers pair up with IRF9, forming IFN-stimulated gene factor 3 (ISGF3), which activate IFN-induced genes (ISGs) after binding to IFN-stimulated response elements (ISREs) located in the ISG promoters. ISGs include genes that encode IL6, IRF7, and components of ISGF3 factors. Augmented synthesis of IRFs 7, 9 and STATs 1, 2 further amplify the effect due to the activation of IFN-β and -α, forming a positive-feedback regulatory loop [32,33]. Mechanistically, RKIP binds to and facilitates TBK1 auto-phosphorylation after viral infection, and significantly, RKIP itself is a substrate of TBK1. It has been shown that phosphorylation of RKIP at Ser^109^ strengthens its interaction with TBK1 [19]. The signaling targets involved in RKIP-mediated enhancement of T cells-dependent systemic inflammatory response caused by exaggerated production of IFN-γ is currently not known.

Although RKIP has been shown as an immune modulator [34,35], the effect of RKIP on immune response genes in breast cancer has never been reported. Our genome-wide microarray results documented for the first-time the effects of an RKIP knockdown on gene expression in cultured breast cancer cells. We showed that a decrease in RKIP expression by specific small interfering RNA increased the expression of genes involved with immune response with the enrichment of genes vital for type 1 and 2 IFN-mediated inflammatory responses in breast cancer cells. The results were substantiated with our meta-analysis of the gene expression database in clinical breast cancer. Our present study therefore provides an experimental proof that RKIP is a potential immune modulator in breast cancer.

While type 1 IFN has five members including IFN-α and -β, type 2 IFN consists of IFN-γ. Unlike type 1 IFN, the expression of type 2 IFN is not directly regulated by IRF but is induced by a number of mitogens and cytokines, particularly IL-2, IL-12, IL-15, IL-18, and type I IFN [36,37]. The logic of why RKIP has an opposite effect on IFN response in macrophages and breast cancer cells is currently unrecognized, and the effect of RKIP on TBK1 in transformed non-immune cells is presently not known. Although TBK1 is the preferred kinase for activation phosphorylation of IRF-3 and -7 in immune cells, IKKε, a close homologue of TBK1, was also identified as a physiological kinase of both IRFs [38]. IKKε was identified as an amplified and overexpressed oncogene by an integrative genomic approach in breast cancer. It remains to be determined if IKK-ε is the predominant driver of IFN response in breast cancer and if it is a target of RKIP. It is also not recognized how IFN-γ signaling is negatively regulated by RKIP in cancer cells. It is possible that RKIP may indirectly regulate IFN type II signaling through type I.

IFN is a class of soluble immune chemical mediators that are characterized by their anti-viral function. Virally infected cells and cancer cells may both be recognized by the immune system. Both fragmented nucleic acids generated by genome unstable cancer cells and viral DNA/RNA stimulate the same pathways through pattern recognition receptors (PRRs) and can illicit an immune response. IFN response is an important component of the cancer surveillance system; its effects on cancer are complex and cancer-type specific. The IFN response enables the host immune system to recognize and eliminate tumorigenic cells. Murine models deficient in type I IFN receptors or II receptors showed increased tumorigenesis compared to wild-type models, suggesting the significance of both classes of IFN in restraining tumor development [39]. Whereas IFN-γ has antitumor effects via directly targeting tumorigenic cells, it was suggested that type I IFN mediates the antitumor effects via stimulation of host hematopoietic cells [39,40]. IFN response has also demonstrated a supportive role for cancer progression and metastasis. Type I IFN response leads to cancer inflammation, resulting in a conducive tumor microenvironment with increased angiogenesis and immune cell infiltration. Type II IFN signaling also has a protumor effect. IFN-γ upregulates T regulatory cells (Treg) and inhibits CD8 T cells [41]. A variety of other mechanisms, including attenuation of neutrophils, upregulation of MHC II, and expression of nonclassical MHC I, have also been implicated to contribute to the tumorigenic effects of IFN-γ [42].

One out of eight women here in the United States will be diagnosed with breast cancer in their lifetime. Each year, an estimated 40,000 women will succumb to breast cancer in the US. IFN-stimulated genes are widely expressed across the different breast cancer subtypes [43]. Significantly, breast cancer tumors expressing higher IFN response gene signatures had higher proclivity to metastasize as compared to tumors expressing low levels of IFN response genes [44]. Here, we showed that the expression of IFN response genes negatively correlated with RKIP expression in clinical breast cancer samples. Furthermore, our results with cell-based studies demonstrated a causal effect of RKIP on the expression of IFN response genes. RKIP has a causal suppressive role in breast cancer metastasis in cancer cell transplantation experimental mouse models. We previously reported that the low RKIP expression levels in breast cancer are associated with a poor prognosis.

## 5. Conclusions

Taken together, our results therefore suggest that RKIP may interfere with breast cancer metastasis by negatively impinging on IFN signaling pathways in breast cancer. At present, it is not clear if the effects of RKIP on IFN signaling pathways are cancer cell-type specific. Identifying targeting IFN signaling as a possible cause of RKIP-mediated suppression of breast cancer metastasis will provide additional drug targets for therapeutic intervention.

## Figures and Tables

**Figure 1 cancers-14-03605-f001:**
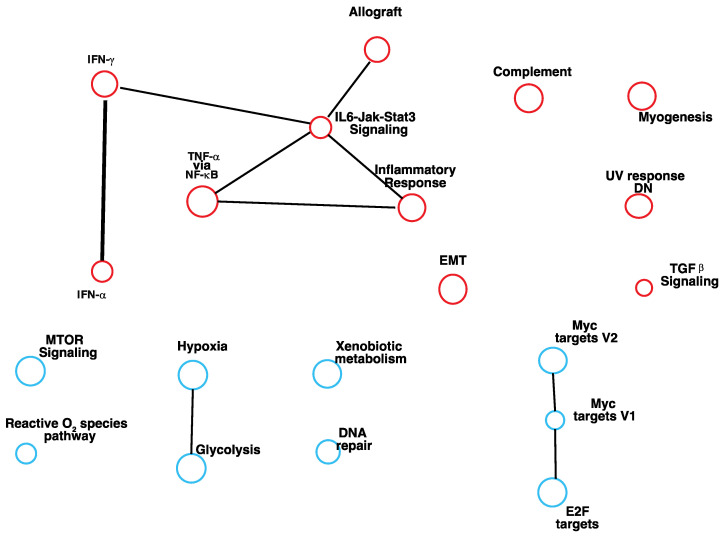
Expression of immune response gene sets correlates with RKIP expression in cultured mouse cancer cells. Cytoscape Enrichment Map with each node (circle) represents a gene set with the indicated biological state. The expression levels of each gene set can be either upregulated (red) or downregulated (blue) in response to reduced RKIP expression. Gene sets with shared genes are linked by edges (dark lines) with thicknesses that correspond to the number of genes in common between two sets. The overlap coefficient was set at 0.2, so that the edges are shown for sets that have about 20% shared genes. Only gene sets with FDR < 0.1 and *p* < 0.03 are shown.

**Figure 2 cancers-14-03605-f002:**
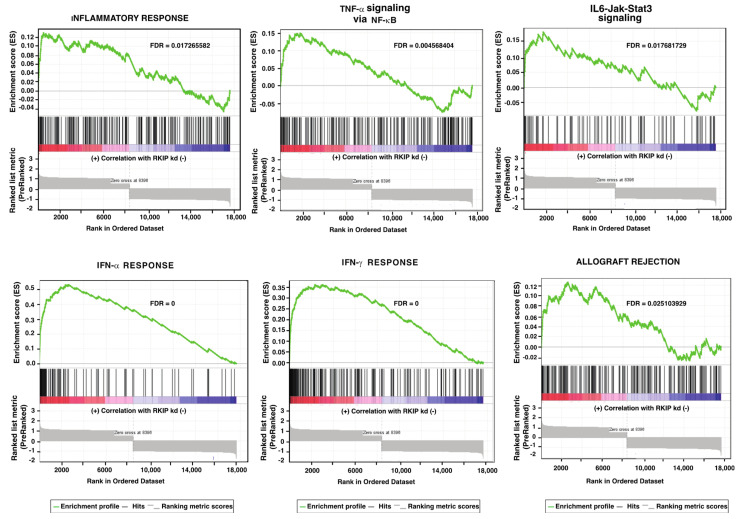
In RKIP knockdown168FARN mouse breast cancer cells, gene sets for immune response were significantly enriched. Enrichment score (ES) are displayed on the y-axis. Vertical black lines on the x-axis represent different genes in the gene sets. Each gene was assigned an ES that was calculated as the maximum deviation from zero as it is going down the ranked list. ES represents the degree of over-representation of a gene set at the top or at the bottom of the ranked gene list. The degree of correlation of genes with the RKIP knockdown are color coded with red for positive and blue for negative correlation. Significance threshold set at FDR < 0.03.

**Figure 3 cancers-14-03605-f003:**
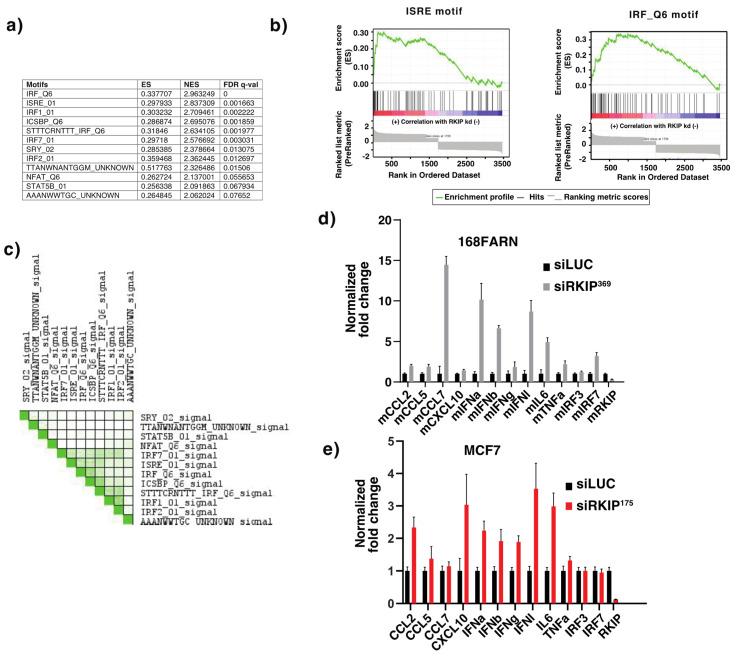
(**a**) The table shows a list of motifs in the MSigDb Transcription Target collection that were enriched in RKIP knockdown168FARN mouse breast cancer cells with FDR < 0.08. (**b**) Enrichment graphs for the top two motifs have the same format as the graphs in Figure 2. (**c**) The set-to-set graph shows the overlap in the leading-edge genes shared by the top 10 gene sets from the GSEA analysis using the TF binding motif gene sets and the si175 v siLuc gene list. Six of the top ten gene sets have many of the leading-edge genes in common, as shown by the dark green squares. Most of these correspond to binding motifs for IRFs (interferon regulatory factors). (**d**) Relative mRNA levels of indicated immune response genes normalized with mGAPDH (mean ± SE), as quantified by qRT-PCR in 168FARN cells expressing the indicated siRNA. The experiments were repeated three times with similar results. (**e**) Relative mRNA levels of indicated immune response genes normalized with GAPDH (mean ± SE), as quantified by qRT-PCR in MCF7 cells expressing indicated siRNA. The experiments were repeated three times with similar results.

**Figure 4 cancers-14-03605-f004:**
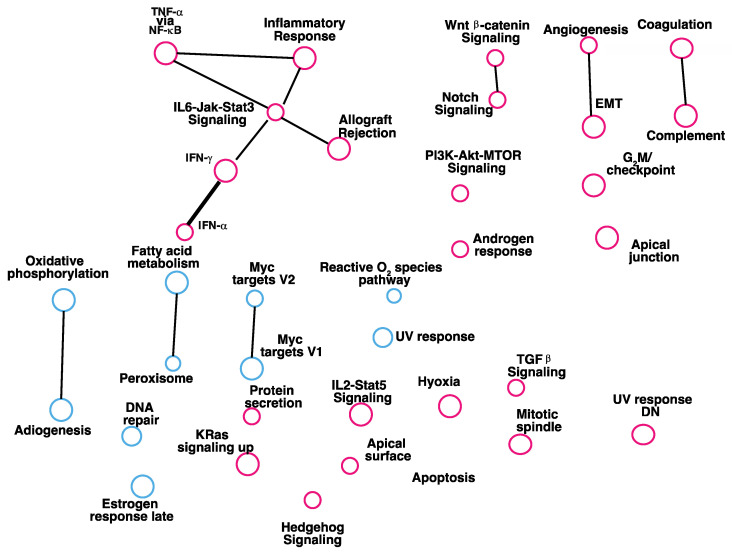
Expression of immune response gene sets correlates with RKIP expression in clinical breast cancers. Cytoscape Enrichment Map with each node (circle) represents a gene set with the indicated biological state. The expression levels of each gene set were either upregulated (red) or downregulated (blue) in low-RKIP-expressing cancers. Gene sets with shared genes are linked by edges ( dark lines) with thicknesses that correspond to the number of genes in common between two sets. Only gene sets with FDR < 0.1 and *p* < 0.03 are shown.

**Figure 5 cancers-14-03605-f005:**
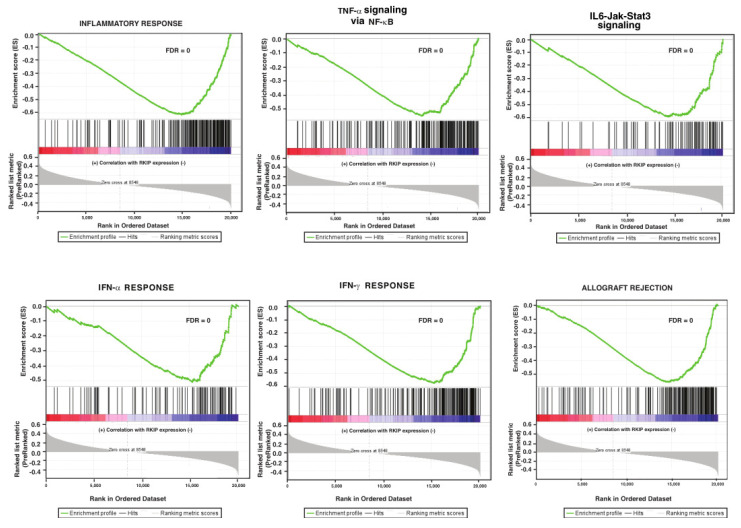
In RKIP low-expressing clinical breast cancer, gene sets for immune response were significantly enriched. Enrichment scores (ES) are displayed on the y-axis. Vertical black lines on the x-axis represent different genes in the gene sets. Each gene was assigned an ES that was calculated as the maximum deviation from zero as it is going down the ranked list. ES represents the degree of over-representation of a gene set at the top or at the bottom of the ranked gene list. The colored band at the bottom represents the degree of correlation of genes with RKIP expression (red for positive and blue for negative correlation). Significance threshold set at FDR < 0.03.

**Table 1 cancers-14-03605-t001:** GSEA kd up and negative correlation. The top hits are listed from the GSEA analysis of the RKIP knockdown (genes upregulated) and show those that are negatively correlated with RKIP expression in the breast cancer TCGA samples. All nine gene sets with q values less than 0.05 are listed. The top 11 ranked by NES of the sets negatively correlated with RKIP expression are listed. The gene signatures shared between the two groups are highlighted. NES, Normalized Enrichment Score.

RKIP kd vs. Control Up	NES	NOM *p*-Value	FDR q-Value	Negative Correlation RKIP	NES	NOM *p*-Value	FDR q-Value
**INTERFERON_ALPHA_RESPONSE**	6.040572	0	0	**INFLAMMATORY_RESPONSE**	−2.80127	0	0
**INTERFERON_GAMMA_RESPONSE**	5.630654	0	0	**EPITHELIAL_MESENCHYMAL_TRANSITION**	−2.6795	0	0
**EPITHELIAL_MESENCHYMAL_TRANSITION**	3.491317	0	0	**INTERFERON_GAMMA_RESPONSE**	−2.62019	0	0
**TNFA_SIGNALING_VIA_NFKB**	2.311845	0	0.004568	**ALLOGRAFT_REJECTION**	−2.52816	0	0
**UV_RESPONSE_DN**	2.304864	0.001919	0.003655	**TNFA_SIGNALING_VIA_NFKB**	−2.49759	0	0
**INFLAMMATORY_RESPONSE**	2.031128	0.005556	0.017266	**UV_RESPONSE_DN**	−2.44412	0	0
**COMPLEMENT**	1.950489	0.006085	0.022258	**KRAS_SIGNALING_UP**	−2.44371	0	0
**ALLOGRAFT_REJECTION**	1.911962	0.006061	0.025104	**IL6_JAK_STAT3_SIGNALING**	−2.39554	0	0
**IL6_JAK_STAT3_SIGNALING**	1.890129	0.017682	0.025151	**COMPLEMENT**	−2.29257	0	0
				**MITOTIC_SPINDLE**	−2.27098	0	0
				**INTERFERON_ALPHA_RESPONSE**	−2.11048	0	0

**Table 2 cancers-14-03605-t002:** GSEA kd down and positive correlation. The top hits are listed from the GSEA analysis of the RKIP knockdown (genes downregulated) and those that are positively correlated with RKIP expression in the breast cancer TCGA samples. All eight gene sets with q values less than 0.2 are listed. The top 10 ranked by NES of the sets positively correlated with RKIP expression are listed. The gene signatures shared between the two groups are highlighted. NES, Normalized Enrichment Score.

RKIP kd vs. Control Down	NES	NOM *p*-Value	FDR q-Value	Positive Correlation RKIP	NES	NOM *p*-Value	FDR q-Value
**MYC_TARGETS_V1**	−3.33323	0	0	**OXIDATIVE_PHOSPHORYLATION**	3.495134	0	0
**E2F_TARGETS**	−2.42757	0	0.002051	**MYC_TARGETS_V1**	2.596274	0	0
**MYC_TARGETS_V2**	−2.24774	0	0.00527	**DNA_REPAIR**	2.395833	0	0
**REACTIVE_OXYGEN_SPECIES**	−2.09321	0.004107	0.013857	**MYC_TARGETS_V2**	2.364108	0	0
**DNA_REPAIR**	−1.83964	0.008949	0.051236	**ADIPOGENESIS**	1.905392	0	1.45 × 10^−4^
**KRAS_SIGNALING_UP**	−1.63499	0.036403	0.12295	**FATTY_ACID_METABOLISM**	1.816652	0	4.45 × 10^−4^
**OXIDATIVE_PHOSPHORYLATION**	−1.51916	0.06993	0.18503	**REACTIVE_OXYGEN_SPECIES**	1.696399	0	0.001005
**ESTROGEN_RESPONSE_EARLY**	−1.50451	0.052863	0.1742	**PEROXISOME**	1.475564	0	0.015246
				**ESTROGEN_RESPONSE_LATE**	1.38756	0	0.033137
				**UV_RESPONSE_UP**	1.293321	0.026144	0.072677

## Data Availability

Expression values for the microarray experiment were deposited to NCBI GEO with the accession number GSE206259.

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
