# Peer review of "A Negative Regulatory Role for RKIP in Breast Cancer Immune Response"

_cancers, 2022, doi:10.3390/cancers14153605_

Round 1

Reviewer 1 Report

I don't have any more comments.

Author Response

No additional comments from reviewer #1 to address. Thank you for taking time to review our revised manuscript.

Reviewer 2 Report

Cancers-1820402

A negative regulatory role for RKIP in breast cancer immune response

This is a revised version of a previously submitted manuscript. The authors were supposed to provide authors’ point-by-point response in the cover letter and the revised areas of the manuscript should have been highlighted. However, the version available to review does not have the changes highlighted. Additionally, no cover letter was provided with author’s response to reviewer’s comments which made it very difficult to assess what changes were made in this version from the original version. The authors should still address the following comments:

 Comments:

1.     This study does not provide mechanistic advances, experimental proof or novelty, since it has already been shown that RKIP is an immune modulator (Gabriela-Freitas  et al., 2019; Zaravinos et al., 2018)

2.     The authors indicate several pathways are modulated by RKIP (Fig 4) but do not assess, validate or find a new pathway by which RKIP is modulating the immune system.

3.     It is not clear if the study is for global breast cancer or a particular subtype. It would be important to analyze the importance in critical subtypes of breast cancer.

4.     Why did the authors choose to use mouse breast cancer cell line for the in vitro experiment and the compare the findings with human breast tumor data sets?

5.     The authors should use multiple cell lines and primary samples to conduct in vitro experiments.

6.     168FARN is a triple negative breast cancer cell line. Did the authors use triple negative breast tumor data sets for the comparison? 

7.     Supplementary Table 1: the authors state “The GSEA results classifies gene sets with positive or negative correlation with PEBP1 expression”. “Conclusion: the negative association of PEBP1 expression with immune function genes holds for all breast cancer subtypes.” Why PEBP1? Why not RKIP?

References

Gabriela-Freitas M, Pinheiro J, Raquel-Cunha A, Cardoso-Carneiro D, Martinho O. RKIP as an Inflammatory and Immune System Modulator: Implications in Cancer. Biomolecules. 2019;9(12):769. Published 2019 Nov 22. doi:10.3390/biom9120769

Zaravinos A, Bonavida B, Chatzaki E, Baritaki S. RKIP: A Key Regulator in Tumor Metastasis Initiation and Resistance to Apoptosis: Therapeutic Targeting and Impact. Cancers (Basel). 2018;10(9):287. Published 2018 Aug 24. doi:10.3390/cancers10090287

Author Response

This is a revised version of a previously submitted manuscript. The authors were supposed to provide authors’ point-by-point response in the cover letter and the revised areas of the manuscript should have been highlighted. However, the version available to review does not have the changes highlighted. Additionally, no cover letter was provided with author’s response to reviewer’s comments which made it very difficult to assess what changes were made in this version from the original version. The authors should still address the following comments:

 We did provide our point-by-point response in the cover letter and a version of manuscript with revised area highlighted. For some reasons the required supporting materials did not make it to the reviewers. The  the marked manuscript is now attached. The response to the comments are listed below.

The response in the attachment covers your comments #1-6. For comment #7

Supplementary Table 1: the authors state “The GSEA results classifies gene sets with positive or negative correlation with PEBP1 expression”. “Conclusion: the negative association of PEBP1 expression with immune function genes holds for all breast cancer subtypes.” Why PEBP1? Why not RKIP?

Response: PEBP1 is the official name of RKIP.

The response to your other comments are also listed below:

This study does not provide mechanistic advances, experimental proof or novelty, since it has already been shown that RKIP is an immune modulator (Gabriela-Freitas et al., 2019; Zaravinos et al., 2018)

Although we agree that RKIP has been shown as an immune modulator, the effect of RKIP on immune response genes in breast cancer has never been reported. Our microarray results are the only genome-wide study of the effect of an RKIP knockdown on gene expression. Our analysis showed an agreement between the effects of the RKIP kd and correlation of RKIP expression in clinical breast cancer. Our present study provides an experimental proof that RKIP is a potential immune modulator in breast cancer.

The authors indicate several pathways are modulated by RKIP (Fig 4) but do not assess, validate or find a new pathway by which RKIP is modulating the immune system.

 New qRT-PCR results have now added to the paper validating several putative pathways that are targeted by RKIP for immune modulation.

 It is not clear if the study is for global breast cancer or a particular subtype. It would be important to analyze the importance in critical subtypes of breast cancer.

We agree that it is important to determine it the effects of RKIP on hallmark signaling pathways are breast cancer subtypes specific. We now include results of expression correlation analysis of RKIP mRNA with other genes by breast cancer subtypes as supplementary materials. Our results showed that the relationship between RKIP expression and identified hallmark pathways is not limited to a specific breast cancer subtypes.

 Why did the authors choose to use mouse breast cancer cell line for the in vitro experiment and the compare the findings with human breast tumor data sets?

 The cancer line 168FARN was chosen because of the existence of an allograft breast cancer model with this cell line for in vivo study and because of its similarity to human breast cancer triple negative subtype. 

The authors should use multiple cell lines and primary samples to conduct in vitro

Two different cell lines have now been used to examine the effects of loss of RKIP on the expression of selected immune response genes by qRT-PCR. The results are shown in Fig. 3g-e

 168FARN is a triple negative breast cancer cell line. Did the authors use triple negative breast tumor data sets for the comparison? 

Triple negative breast tumor data sets have now been added to the comparison. The results were shown in supplementary Table 1 and Figure 1.

 The authors need to provide experimental proof for their findings. The authors should have performed in vitro experiments to validate some of the genes that are regulated by RKIP downregulation, such as Western blotting or RT-PCR. 

 In vitro experiments have now been performed to validate selected RKIP regulated genes identified by DNA microarray analysis by qRT-PCR. The results are shown in Figure 3d-e.

This study lacks in vivo confirmation of any of the targets identified by bioinformatics.

In vivo confirmation of identified targets with mouse cancer models will be the scope of future studies.

Figure font size is inconsistent and often illegible.

Figure font size has been adjusted for consistency and legibility.

Round 2

Reviewer 2 Report

Cancers-1820402

A negative regulatory role for RKIP in breast cancer immune response

This is a revised version of a previously submitted manuscript. The authors attempted to address the previous comments with additional experiments however, the authors should still address the following comments:

 Comments:

1.     This study does not provide mechanistic advances, experimental proof or novelty, since it has already been shown that RKIP is an immune modulator (Gabriela-Freitas  et al., 2019; Zaravinos et al., 2018)

The authors added some comments in the response letter regarding this, which should be added in the paper.

2.     Why did the authors choose to use mouse breast cancer cell line for the in vitro experiment and the compare the findings with human breast tumor data sets?

The authors added some clarification in the response letter, which should be added in the paper.

3.     The authors should use multiple cell lines and primary samples to conduct in vitro experiments.

Authors examined the effects of loss of RKIP on the expression of selected immune response genes by qRT-PCR in MCF7 and 168FARN cells (3g and e). The authors should add more human breast cancer cell line preferably TNBC to further strengthen the observation.

4.     Supplementary Table 1: It’s better to write PEBP1 (=RKIP) instead of PEBP1 only.

Author Response

  1. This study does not provide mechanistic advances, experimental proof or novelty, since it has already been shown that RKIP is an immune modulator (Gabriela-Freitas et al., 2019; Zaravinos et al., 2018)

The authors added some comments in the response letter regarding this, which should be added in the paper.

Response: Thanks for the suggestion. The response to your comments are now added in the Discussion section of the paper. 

  1. Why did the authors choose to use mouse breast cancer cell line for the in vitro experiment and the compare the findings with human breast tumor data sets?

The authors added some clarification in the response letter, which should be added in the paper.

Response: Thank you. The clarifications are now added in the Discussion section of the paper. 

  1. The authors should use multiple cell lines and primary samples to conduct in vitro experiments.

Authors examined the effects of loss of RKIP on the expression of selected immune response genes by qRT-PCR in MCF7 and 168FARN cells (3g and e). The authors should add more human breast cancer cell line preferably TNBC to further strengthen the observation.

Response: we did examine the effect of RKIP silencing on the expression of the same set of immune response genes in TNBC BT20 cells. We observed similar trend in the effect of gene expression upon decreased in RKIP expression. However the differences were comparatively small compared with results generated  with MCF7 and 168FARN cells (3g and e) and were presented in this paper. 

  1. Supplementary Table 1: It’s better to write PEBP1 (=RKIP) instead of PEBP1 only.

Response: thank you for the suggestion. PEBP1 was re-written as PEBP1/RKIP in supple Table 1.

This manuscript is a resubmission of an earlier submission. The following is a list of the peer review reports and author responses from that submission.

Round 1

Reviewer 1 Report

In this manuscript, Vu N. Bach et al. identified a possible negative regulatory role for RKIP mainly by using Gene set enrichment analysis (GSEA) for microarray profiling data in ctrl and RKIP know-down FARN168 cell line. Generally speaking, the data provided by the authors are very limited, most of the conclusions were also based on insufficient evidence. I suggest that the authors should take more breast cancer cell lines, at least one or two human breast cancer cell line, performed more comprehensive analysis for microarray profiling data or RNA-seq data (differential expression analysis, GO analysis, TF regulatory networks of the DEGs…), randomly selected some DEGs and confirm the gene expression using RT-qPCR and western blot, at least the affected genes in immune response. Submit your microarray data to NCBI GEO data. This manuscript was also not well-written, the author should spend more time to edit, many grammar and punctuation errors were easily observed. 

Reviewer 2 Report

cancers-1415903

Identify a negative regulatory role for RKIP in breast cancer immune response

In this manuscript, the authors demonstrate that RKIP expression is associated with repression of signaling pathways and genes involved in immune response in breast cancer cells based on in vitro data and bioinformatics-driven approach and discuss its potential ramifications on cancer therapy. The authors state that the RKIP effect on IFN signaling pathways may be cancer cell type specific and can be used as an additional drug target for therapeutic intervention of triple negative breast cancer (TNBC).

Comments:

  1. ThIS study does not provide mechanistic advances, experimental proof or novelty, since it has already been shown that RKIP is an immune modulator (Gabriela-Freitas et al., 2019; Zaravinos et al., 2018)
  2. The authors indicate several pathways are modulated by RKIP (Fig 4) but do not assess, validate or find a new pathway by which RKIP is modulating the immune system.
  3. It is not clear if the study is for global breast cancer or a particular subtype. It would be important to analyze the importance in critical subtypes of breast cancer.
  4. Why did the authors choose to use mouse breast cancer cell line for the in vitro experiment and the compare the findings with human breast tumor data sets?
  5. The authors should use multiple cell lines and primary samples to conduct in vitro
  6. 168FARN is a triple negative breast cancer cell line. Did the authors use triple negative breast tumor data sets for the comparison? 
  7. The authors need to provide experimental proof for their findings. The authors should have performed in vitro experiments to validate some of the genes that are regulated by RKIP downregulation, such as Western blotting or RT-PCR. 
  8. This study lacks in vivo confirmation of any of the targets identified by bioinformatics.
  9. Figure font size is inconsistent and often illegible.

References

Gabriela-Freitas M, Pinheiro J, Raquel-Cunha A, Cardoso-Carneiro D, Martinho O. RKIP as an Inflammatory and Immune System Modulator: Implications in Cancer. Biomolecules. 2019;9(12):769. Published 2019 Nov 22. doi:10.3390/biom9120769

Zaravinos A, Bonavida B, Chatzaki E, Baritaki S. RKIP: A Key Regulator in Tumor Metastasis Initiation and Resistance to Apoptosis: Therapeutic Targeting and Impact. Cancers (Basel). 2018;10(9):287. Published 2018 Aug 24. doi:10.3390/cancers10090287